# Genome-Wide Expression and Anti-Proliferative Effects of Electric Field Therapy on Pediatric and Adult Brain Tumors

**DOI:** 10.3390/ijms23041982

**Published:** 2022-02-11

**Authors:** Joshua Branter, Maria Estevez-Cebrero, Mohammed Diksin, Michaela Griffin, Marcos Castellanos-Uribe, Sean May, Ruman Rahman, Richard Grundy, Surajit Basu, Stuart Smith

**Affiliations:** 1Children’s Brain Tumour Research Centre, School of Medicine, University of Nottingham, Nottingham NG7 2UH, UK; joshuabranter@outlook.com (J.B.); mszme1@exmail.nottingham.ac.uk (M.E.-C.); msxmad@exmail.nottingham.ac.uk (M.D.); michaela.griffin@nottingham.ac.uk (M.G.); ruman.rahman@nottingham.ac.uk (R.R.); richard.grundy@nottingham.ac.uk (R.G.); 2Nottingham Arabidopsis Stock Centre, Division of Plant and Crop Sciences, School of Biosciences, University of Nottingham, Loughborough LE12 5RD, UK; sbzmc3@exmail.nottingham.ac.uk (M.C.-U.); sean.may@nottingham.ac.uk (S.M.); 3Department of Neurosurgery, Nottingham University Hospitals, Nottingham NG7 2UH, UK; Surajit.Basu@nuh.nhs.uk

**Keywords:** glioma, ependymoma, tumor treating fields, Deep Brain Stimulation

## Abstract

The lack of treatment options for high-grade brain tumors has led to searches for alternative therapeutic modalities. Electrical field therapy is one such area. The Optune™ system is an FDA-approved novel device that delivers continuous alternating electric fields (tumor treating fields—TTFields) to the patient for the treatment of primary and recurrent Glioblastoma multiforme (GBM). Various mechanisms have been proposed to explain the effects of TTFields and other electrical therapies. Here, we present the first study of genome-wide expression of electrotherapy (delivered via TTFields or Deep Brain Stimulation (DBS)) on brain tumor cell lines. The effects of electric fields were assessed through gene expression arrays and combinational effects with chemotherapies. We observed that both DBS and TTFields significantly affected brain tumor cell line viability, with DBS promoting G0-phase accumulation and TTFields promoting G2-phase accumulation. Both treatments may be used to augment the efficacy of chemotherapy in vitro. Genome-wide expression assessment demonstrated significant overlap between the different electrical treatments, suggesting novel interactions with mitochondrial functioning and promoting endoplasmic reticulum stress. We demonstrate the in vitro efficacy of electric fields against adult and pediatric high-grade brain tumors and elucidate potential mechanisms of action for future study.

## 1. Introduction

Glioblastoma multiforme (GBM) is the most common and aggressive adult primary brain tumor [1]. The current standard of care is radical surgical resection along with chemoradiation therapy with temozolomide (TMZ) [2]. Despite decades of advancement in the laboratory with regards to potential treatments, little progress has been made in the clinic as GBM is still an exceptionally poor prognosis tumor. More recently, there is growing pre-clinical [3,4,5,6] and clinical data [3,4] suggesting that electric fields may present an effective treatment modality for high-grade brain tumor patients. It is proposed in existing studies that electric fields may function through perturbation of polarized cytokinetic proteins [5,6], disruption in membrane permeability [7] and promotion of autophagy [8], but further emerging mechanisms are currently being elucidated.

Recently, exciting work has demonstrated that glioma cells may rely on functional synaptic connections with both neurons and other tumor cells [9,10]. Potassium currents evoked by these electrical mechanisms form an intra-tumoral electrical network, and blocking this network pharmacologically can limit glioma growth. This work highlights the importance of electrophysiology in GBM, and its manipulation (by exogenous current) may hold promise as a therapeutic avenue for treating these incurable cancers. The exact mechanism for the effects of electrical fields on cancer cells has implications for the future development and optimization of this emerging therapeutic modality.

At present, the only clinical electrotherapy approved for adult GBM is the Optune™ system (Novocure Inc. St Helier, Jersey). The system includes a portable electric field generator which delivers low intensity (1–3 V/cm), intermediate frequency (200 kHz) and alternating electric fields—termed Tumor Treating Fields (TTFields) [4,6]. The TTFields are delivered to the patient’s brain via external electrode arrays adhered to the scalp. TTFields therapy has been evaluated in multiple phase III trials, although some criticisms have been made of trial design [11]. Published studies suggest benefit to both recurrent [3] and primary GBM patients [4]. Concerns have been raised regarding therapeutically relevant intensities of TTFields at the initial tumor site [12].

Development of internalized electrotherapies seek to exploit the electrosensitivity of tumor cells while overcoming limitations of delivery from an external system, as well as overcoming concerns of cost of the TTFields therapy [13]. Xu and colleagues [14] described their preliminary in vitro investigation of an internalized electrode system, with their key finding being that 4 V/130 Hz electric fields induced apoptosis in primary GBM cells and have no notable effect on primary neurons, and efficacy may be improved with the addition of TMZ. More recently, the internalized approach (termed Intratumoral Modulation Therapy—IMT) was used to deliver 200 kHz electric fields in in vitro and in vivo models [15]. The main findings of this study were in line with the previous reports, demonstrating efficacy against GBM cells while not affecting primary neurons, as well as showing greater efficacy when combined with TMZ. Notably, the study also demonstrated the safety of IMT in vivo as well as significant reductions in tumor volume across their pilot 15-animal cohort study.

Electrical fields have been delivered clinically to the human brain for many years via Deep Brain Stimulating (DBS) electrodes. In contrast to TTFields, DBS are internalized wires implanted surgically into the brain parenchyma. They have shown considerable success in treating movement disorders such as Parkinson’s disease, through delivery of electrical fields to designated anatomical targets such as the subthalamic nucleus or globus pallidus. Many thousands of patients have benefitted from DBS, and the electrode system is well-tolerated, internalized and safe to implant. IMT electrodes, in contrast, have not yet been used in humans. The electrical field delivered by DBS also differs from TTFields in frequency, generally being in the hundreds of hertz range, as opposed to the hundreds of kilohertz for TTFields.

Here, we present our investigations into the effects of TTFields on pediatric brain tumor cell lines as well as comparative studies into the use of repurposed DBS electrodes as delivery devices for low frequency electric fields. We demonstrate how electric field treatments of varying intensities and frequencies delivered by either TTFields or DBS electrodes significantly affect primary and commercial brain tumor cell line viability and cell cycling, while not affecting the viability of non-dividing astrocyte cells. We also show how efficacy of electric fields, including the clinically relevant TTFields (Optune™), may be significantly improved with the addition of TMZ and mitotic inhibitors. Finally, we present genome-wide expression analysis of electric field-treated GBM cells, examining the genetic effects of both TTFields and DBS treatments on tumor cells, giving further suggestions to possible mechanisms of action.

## 2. Results

### 2.1. Tumor Treating Fields Demonstrate Efficacy That Varies in a Frequency-Dependent Manner

TTFields were delivered in vitro using the Inovitro laboratory testing system that replicates the effect of the clinically used technique (the Optune™ device). We determined the optimum frequency of TTFields for our cell lines over a panel of frequencies. TTFields have been tested in various cancer cell lines within the 100–500 kHz range [6], and thus far, TTFields have been used clinically on adult GBM patients at a frequency of 200 kHz [4]. We ran a panel of pediatric GBM, medulloblastoma and ependymoma cell lines and a primary adult invasive margin GBM cell line over a 100–400 kHz TTFields range for a 72 h treatment. We then measured cell viability via PrestoBlue, and we validated the relation of PrestoBlue to cell viability through cell counts. We confirmed that all our cell lines were sensitive to TTFields treatment across the frequencies tested. The GBM cell lines KNS42, SF188 and GIN-31 had an optimal frequency of 200 kHz, 400 kHz and 200 kHz, respectively, while experiencing a reduction of 61%, 50% and 40% in the cell count (*p* ≤ 0.0001; *t*-test). The medulloblastoma cell lines, DAOY and UW228-3, had an optimal frequency of 300 kHz and 100 kHz, with reduction of 30% and 47% in the cell counts (*p* ≤ 0.0001; *t*-test). The ependymoma cell lines tested, DKFZ-EPN1 and BXD-1425EPN, had optimal frequencies of 100 kHz and 200 kHz, with reductions of cell counts of 60% and 40% (*p* ≤ 0.0001; *t*-test). Interestingly, the pediatric GBM cell line SF188 was determined to have an optimal frequency of 400 kHz as opposed to the accepted 200 kHz for adult GBM in the clinic; however, KNS42 cells were most affected by TTFields at 200 kHz (Figure 1).

We sought to confirm the effects of TTFields treatment on non-neoplastic (but still mitotically active) neural stem cells and post-mitotic astrocytes. We treated both types of cell for 72 h at 200 kHz and 400 kHz TTFields, followed by a PrestoBlue assay for assessment of cell viability and a cell count relative to the control cells. Cell counts confirmed our hypothesis that TTFields would only negatively affect the actively dividing cells, with a reduction in neural stem cell numbers but no significant effect on astrocyte numbers. The reduction of human neural stem cell counts following TTFields treatment were 41% and 37% (*p* = 0.0018 and *p* = 0.0024; *t*-test) following treatment with 200 kHz and 400 kHz TTFields, respectively. There were no significant differences between the 200 kHz and 400 kHz treatments (Figure 1C).

### 2.2. Electric Fields Delivered from DBS Reduce Metabolic Viability in a Voltage- and Frequency-Dependent Manner in GBM Cell Lines

First, we determined the optimal parameters for DBS electric fields treatment. We used clinically available DBS wires identical to those used for DBS in Parkinson’s disease patients. We ran a panel of voltages, 0 V, 0.5 V, 1.0 V, 2.5 V, 5.0 V and 10.0 V, against a representative cohort of GBM cell lines, U87, GIN-5, GIN-31 and KNS42. These voltages covered the range of intensities which could be technically achieved from the pulse generators used to generate the electrical field, at a frequency of 130 Hz and a pulse width of 450 µs—a frequency chosen as to its clinical relevance with neuromodulation therapy [16]. GBM cell lines were treated with DBS electric fields for 7 days with their metabolic activity being measured intermittently at Day 0, 1, 5 and 7 in order to determine the field intensity which had the greatest effect on the cell lines, as well as addressing the effect of treatment duration on the efficacy of DBS electric fields. Higher intensities as well as a longer duration of treatment were correlated with the greatest reduction in metabolic activity at the endpoint across all cell lines. The reductions in viability following a 7-day treatment at 10.0 V/130 Hz/450 µS were: 65% (*p* ≤ 0.0001, two-way ANOVA) for U87 cells, 69% (*p* ≤ 0.0001, two-way ANOVA) for KNS42 cells, 65% (*p* ≤ 0.0001, two-way ANOVA) for SF188 cells, 39% (*p* ≤ 0.0001, two-way ANOVA) for GIN-5 cells and 30% (*p* ≤ 0.0001, two-way ANOVA) for GIN-31 cells (Figure 2A).

We investigated the influence of DBS frequency on cell viability at an intensity shown to be effective. The panel of frequencies selected was 60 Hz, 90 Hz, 130 Hz, 160 Hz, 190 Hz, 500 Hz and 1000 Hz (Figure 2B). Cells were again treated for 7 days, with the 60–190 Hz frequencies run at 5.0 V, as this intensity was the highest possible programmed voltage from the pulse generators used. Frequencies bordering on the limits of being clinically relevant, 60 Hz and 190 Hz, were correlated with the greatest reduction in metabolic activity at endpoint across the U87 (60%, *p* ≤ 0.0001 and 65%, *p* ≤ 0.0001, respectively) and KNS42 (58%, *p* ≤ 0.0001 and 71%, *p* ≤ 0.0001, respectively) cell lines. Relative to treatment with 130 Hz, these reductions translate to a further reduction in U87 (15% and 20%, respectively, however did not achieve significance) and KNS42 (19%, *p* = 0.0081 and 32%, *p* ≤ 0.0001) (Figure 2B).

Following showing efficacy of DBS electric fields on our panel of GBM cell lines, we assessed the effect of electric fields on non-dividing astrocytes to determine whether an actively dividing cell was a necessity for effective electrical treatment. We derived non-dividing astrocytes from H1-hESC cells and subjected the cells to a 7-day DBS electric treatment at 10.0 V/130 Hz/450 µS pulse-width, followed by a PrestoBlue assay for assessment of cell viability and a cell count relative to the control cells. Cell counts showed that electric fields did not prove to be cytotoxic to the non-dividing astrocytes. We also performed the same experiment with dividing neural stem cells, which were also differentiated from H1-hESC cells. The cell viability measurements showed a reduction of cell count of 59% (*p* ≤ 0.0001; *t*-test) when treated with DBS electric fields. There were no significant differences between the 200 kHz and 400 kHz treatments (Figure 2C).

### 2.3. Electric Fields Perturb Cell Cycling of Brain Tumor Cell Lines in a Frequency-Dependent Manner

Given that reductions in cell viability were achieved in all cell lines with electrical treatment, we sought to further characterize this phenomenon. Cells were treated with DBS electric fields at 10.0 V/130 Hz/450 µS for 7 days and then their cell cycle profiles were analyzed using propidium iodide (PI) staining and flow cytometry. Firstly, DBS treatment appeared to have minimal cytotoxicity activity as shown through the lack of cell accumulation in the subG0-phase; however, U87 cell accumulation did achieve significance (*p* = 0.0019, two-way ANOVA). Secondly, there was significant G0-phase accumulation following DBS treatment for U87, KNS42 and SF188 cells, with an increase of 15% (*p* ≤ 0.0001, two-way ANOVA), 14% (*p* ≤ 0.0001, two-way ANOVA) and 17% (*p* ≤ 0.0001, two-way ANOVA), respectively. Thirdly, S-phase depletion following DBS treatment was apparent in our cell lines U87, KNS42 and SF188, with a decrease of 13% (*p* ≤ 0.0001, two-way ANOVA), 10% (*p* ≤ 0.0001, two-way ANOVA) and 18% (*p* ≤ 0.0001, two-way ANOVA), respectively. Finally, there was no significant effect on the G2-phase of the cell cycle following DBS treatment for any of our cell lines (Figure 3A).

TTFields have previously been shown to promote G2/M-phase accumulation through interactions with tubulin dynamics [5,17,18]. The panel of pediatric brain tumor cell lines were treated with TTFields for 72 h at their optimal frequencies, and their cell cycling was analyzed using PI staining and flow cytometry. TTFields treatment caused G2/M-phase accumulation for all cell lines, with an increase of 9% and 11% (*p* = 0.0297 and *p* = 0.0037, two-way ANOVA) for KNS42 and SF188 cell lines, 11% and 12% (*p* ≤ 0.0001, two-way ANOVA) for DAOY and UW228-3 cell lines and 13% and 8% (*p* = 0.0040 and *p* = 0.0024, two-way ANOVA) for BXD-1425EPN and DKFZ-EPN1 cell lines, respectively. The cell lines DAOY, UW228-3 and DKFZ-EPN1 also had significant G0-phase depletion at 8%, 7% and 7% (*p* = 0.0002, *p* ≤ 0.0001 and *p* = 0.0057, two-way ANOVA), respectively. These cell lines also experienced a slight S-phase accumulation as well: 6%, 4% and 10% (*p* = 0.0131, *p* ≤ 0.0001 and *p* ≤ 0.0001, two-way ANOVA). None of the cell lines had significant levels of subG0-phase accumulation, suggesting that TTFields has a predominantly cytostatic over a cytotoxic effect in the tested pediatric brain tumor cell lines (Figure 3B).

### 2.4. The Efficacy of Electric Fields Appears to Be Increased with the Addition of Mitotic Inhibitors in GBM Cell Lines

After we established that DBS treatment promoted G0-phase accumulation, we looked to further augment this with combinations of mitotic inhibitors. The mitotic inhibitors used were paclitaxel and mebendazole as they both targeted the latter stages of the cell cycle, the G2/M phase; this strategy allows targeting of multiple stages of the cell cycle for greater coverage. We first established the IC50 of paclitaxel (Appendix A) on the cell lines and then used low dose and high dose paclitaxel on GBM cell lines as a monotherapy and in combination with DBS treatment for 5 days for comparison. We first confirmed that the cell lines were sensitive to the mitotic inhibitors used alone, with reductions of cell viability, compared to no treatment, of 53% (low dose paclitaxel) and 70% (high dose paclitaxel) (*p* ≤ 0.0001, two-way ANOVA) for U87 cells, 61% and 69% (*p* ≤ 0.0001, two-way ANOVA) for KNS42 cells and 39% and 65% (*p* = 0.0037 and *p* ≤ 0.0001; two-way ANOVA) for SF188 cells, respectively (Figure 4A). Comparing the lower and higher doses of paclitaxel in combination with the additive effect of DBS treatment at 10.0 V/130 Hz/450 µS to paclitaxel treated alone, additional reductions in cell viability were recorded in U87 cells at 72% and 32% (*p* ≤ 0.0001 and *p* = 0.0005, two-way ANOVA), KNS42 cells at 58% and 17% (*p* ≤ 0.0001 and *p* = 0.001, two-way ANOVA) and SF188 cells at 67% and 43% (*p* ≤ 0.0001, two-way ANOVA), respectively (Figure 4A).

Following previous reports of the combination of TTFields and mitotic inhibitors promoting apoptosis and greater accumulation of cells in the G2/M-phase of the cell cycle, we sought to investigate how DBS electric fields with the addition of paclitaxel might also affect cells. We analyzed the cell cycle profile of a lower and higher dose of paclitaxel with DBS treatment at 10.0 V/130 Hz/450 µS. Firstly, combining DBS with a higher dose of paclitaxel appeared to create a significant increase in subG0-phase accumulation relative to either treatments alone, with an increase of 13% (*p* ≤ 0.0001, two-way ANOVA) in U87 cells, 9% (*p* = 0.0035, two-way ANOVA) in KNS42 cells and 10% (*p* = 0.0009, two-way ANOVA) in SF188 cells and when compared to cell lines treated with high dose paclitaxel alone. This combination did also cause a significant depletion of G0-phase accumulation, with no significant differences in accumulation in the G2/M-phase for our cells when compared to DBS-treated cells alone. Finally, combining the lower dose of paclitaxel with DBS treatment has no significant effect on the cell cycle (Figure 4B).

Overall, GBM cell lines were all affected by DBS electric fields and paclitaxel as monotherapies and in combination, with the addition of a higher dose paclitaxel promoting significant increases in apoptosis.

### 2.5. The Efficacy of TTFields and DBS Is Increased with the Addition of Chemotherapies in Brain Tumor Cell Lines

As our data suggests that TTFields target the G2/M-phase of the cell cycle, we decided to further target this phase with the tubulin-interacting mitotic inhibitors, paclitaxel and mebendazole. This strategy of targeting the same region of the cell cycle with different therapies seeks to overcome the axis-dependent nature of TTFields which may limit efficacy of the treatment. We established the IC50 of paclitaxel (0.4–4.6 nM) and mebendazole (0.02–1.2 µM) (Appendix A) on various brain tumor cell lines and then took a lower and a higher dose and treated our cell lines as a monotherapy and in combination with TTFields for a 72 h treatment for comparison. We first confirmed that all cell lines were sensitive to the mitotic inhibitors used, with reductions of cell viability of 30% and 39% for low dose paclitaxel and low dose mebendazole, respectively, (*p* = 0.0034 and *p* ≤ 0.0001, two-way ANOVA) for SF188 cells, 32% and 28% (*p* ≤ 0.0001, two-way ANOVA) for UW228-3 cells and 23% and 30% (*p* ≤ 0.0001, two-way ANOVA) for BXD-1425EPN cells.

Comparing the lower and higher doses of paclitaxel in combination with TTFields treatment at the determined optimal frequency to paclitaxel treated alone, reductions in cell viability were recorded in SF188 cells at 37% (low dose plus TTF) and 33% (high dose plus TTF) (*p* ≤ 0.0002 and *p* = 0.0006, two-way ANOVA), UW228-3 cells at 34% and 22% (*p* ≤ 0.0001 and *p* = 0.0012, two-way ANOVA) and BXD-1425EPN cells at 27% and 34% (*p* ≤ 0.0001, two-way ANOVA) (Figure 5A). We also combined TTFields with the mitotic inhibitor, mebendazole. Comparing the lower and higher doses of mebendazole in combination with TTFields treatment at the determined optimal frequency to mebendazole treated alone, reductions in cell viability were recorded in SF188 cells at 40% and 22% (*p* ≤ 0.0001 and *p* = 0.0092, two-way ANOVA), UW228-3 cells at 25% and 29% (*p* ≤ 0.0001 and *p* = 0.0012, two-way ANOVA) and BXD-1425EPN cells at 19% and 16% (*p* ≤ 0.0001, two-way ANOVA) (Figure 5C). Overall, the pediatric brain tumor cell lines were more significantly affected by the combinational approach. The additive effect of TTFields to mitotic inhibitors in cell lines is in line with previous reports [19].

Using the same combination strategy, we also looked to see how combinations of DBS with mebendazole affected brain tumor cells. Comparing the lower and higher doses of mebendazole in combination with DBS treatment at 10.0 V/130 Hz/450 µS to mebendazole treated alone, additional reductions in cell viability were recorded in U87 cells at 53% and 8% (*p* ≤ 0.0001 and *p* = 0.003, two-way ANOVA), KNS42 cells at 42% and 9% (*p* ≤ 0.0001 and *p* = 0.0008, two-way ANOVA), SF188 cells at 49% and 21% (*p* ≤ 0.0001 and *p* = 0.0002, two-way ANOVA) and GIN-31 cells at 17% and 9% (*p* ≤ 0.0001 and *p* = 0.0005, two-way ANOVA), respectively (Figure 5B).

We sought to combine DBS treatment with TMZ, as this may be more clinically relevant to GBM patients and is already the clinical strategy for Optune™. We used multiple cell lines, including MGMT-methylated (U87 and GCE-77) and MGMT-unmethylated (GIN-28 and GIN-31) cell lines, to be more representative of the clinical challenges of GBM treatment. We first established an IC50 of TMZ (0.8–1.2 µM) (Appendix A) on GBM cell lines and then took higher and lower doses and treated our cell lines as a monotherapy and in combination with DBS treatment for 5 days to form a comparison. Comparing the higher and lower doses of TMZ in combination with DBS at 10.0 V/130 Hz/450 µS to TMZ treated alone, reductions in cell viability were recorded in U87 cells of 32% and 33% (*p* ≤ 0.0001, two-way ANOVA), GCE-77 cells at 22% and 29% (*p* ≤ 0.0001, two-way ANOVA), GIN-28 cells at 2% and 5% (*p* ≥ 0.05 and *p* ≤ 0.0001, two-way ANOVA), and GIN-31 cells at 12% and 7% (*p* ≤ 0.0001 and *p* = 0.002, two-way ANOVA), respectively (Figure 5D).

Overall, brain tumor cell lines all showed additional reduction in viability from the addition of electric fields to chemotherapy. Interestingly, the MGMT-unmethylated cell lines (GIN-31 and GIN-28) both benefited from additional electric field treatment with the TMZ, though to a much more modest degree than with methylated cell lines.

### 2.6. Electric Field Treatments Cause Gene Expression Changes in GBM Cell Lines

To further explore alternative potential mechanisms of electric fields as a whole, we ran TTFields or DBS-treated and control KNS42 and GIN-31 cell lines on a genome-wide mRNA expression array (Clariom™ S Human Assays) to produce genome-wide profile of expression for all genes. Hierarchical clustering analysis showed differential expression profiles of our cell lines. Firstly, we observed that DBS electric fields treatment caused differential expression compared to untreated controls of 4539 and 5406 genes for KNS42 and GIN-31 cells, respectively (Appendix A). Of these genes, 3345 were commonly differentially expressed between them (Figure 6A). Similarly, TTFields treatment caused differential expression of 4746 and 5500 genes for KNS42 and GIN-31 cells, respectively, with 3469 commonly differentially expressed genes (Figure 6B).

Gene ontology analysis implicated that the commonly differentially expressed genes were related to genes involved in mitochondrial and ER functioning, including, electron transport, metabolism, ion signaling and protein folding. It is worth noting that all of these associated genes are significantly downregulated, implying that these processes are also downregulated. PrestoBlue assay changes are significantly dependent on mitochondrial function and show substantial decreases as illustrated in previous results sections. Furthermore, KEGG pathway analysis demonstrated significant over-representation of genes known to be involved in the Parkinson, Alzheimer and Huntington disease pathways in our data (Appendix A). Although these diseases have their own pathologies, they share similarities with regards to their causative factors [20]. All three of the aforementioned diseases have been shown to have significant downregulation of NADH dehydrogenase, among other oxidative phosphorylation complexes, and we observed significant downregulation of MT-ND3, MT-ND5 and MT-ND6 (subunits of the NADH dehydrogenase [21]) in our cell lines following electrical treatment.

Finally, we chose MT-ND5, CTSB and BTNL9 to validate the gene expression findings at a protein level via Western blot; of note, MT-ND5 was the most downregulated gene (>−100-fold downregulation) and BTNL9 was the only upregulated gene of the top 50 target genes (Appendix A). Our Western blot importantly confirmed that the expression patterns observed with the gene expression data (mRNA level) are replicated as changes at the protein level (Figure 6C).

## 3. Discussion

We have shown that both electrical treatments (TTFields and DBS) exhibit efficacy against adult and pediatric brain tumor cell lines in vitro, and this efficacy increased with optimization of frequency and other parameters. DBS electric field efficacy was increased through increasing the intensity of the field and through manipulation of frequency between 60 Hz and 190 Hz. We tested frequencies approaching the threshold that is clinically relevant for DBS purposes (~130 Hz) [22] to test whether the effects of DBS electric fields were frequency-specific and found effects at frequencies that are feasible to deliver with existing technology and should not cause unintentional CNS stimulation. Stimulation at lower frequency than 100 Hz would have the potential for stimulating neuronal activity and inadvertent neurological side effects.

TTFields treatment efficacy was increased through manipulation of the frequency between 100 kHz and 400 kHz; we did not explore the effects of manipulating the intensity of TTFields as this has already been established in vitro [23]. We found that TTFields in the range of 100–400 kHz over a 72 h treatment period demonstrated efficacy across all tested cell lines, in line with data suggesting efficacy against actively dividing cells regardless of cell type [17,23,24]. It has also been observed that the optimal frequency determined for cell lines is consistent for tumors of the same origin [23]. However, we show how cell lines each have a clear optimal frequency despite being of the same tumor type, apart from the ependymoma cell lines which more interestingly appear to be equally influenced throughout the whole range of 100–400 kHz TTFields. We also demonstrated a variable optimal frequency for the panel of GBM cell lines. This highlights the possible need for further development of strategies for patient optimization of TTFields treatment. Observations of the optimal frequencies and the corresponding cell size broadly aligns with the hypothesis that the optimal frequency is inversely proportional to the cell size [6,25]. It is possible that future clinical use of the Optune™ device could be improved by analysis of the cell size for the individual patient’s tumor, and a personalized TTFields ‘prescription’ can be developed based on in vitro analysis of optimal electrical field frequency for the individual cells biopsied at surgery.

Previous reports have hypothesized that an actively dividing cell is the target of electric field treatment [17], and a few reports have shown how these treatments have little to no efficacy on non-dividing cells [14,15,17,26]. Our data demonstrates how significantly lower frequency DBS electric field treatment at 10.0 V/130 Hz/450 µS as well as TTFields treatment at 200 kHz and 400 kHz had no effect on cell viability on post-mitotic astrocytes derived from neural stem cells, whereas the treatment still had a significant effect on the dividing neural stem cell line. Given that our astrocytes were derived from the neural stem cell line, these data further demonstrate how DBS electrical treatment targets actively dividing cells, not just cancer cells specifically. Most neurologically functional cells in the central nervous system are not mitotically active (though a potentially active pool of stem-like cells is now generally accepted), and the lack of toxicity against non-dividing cells for electrical fields is encouraging for future translation and clinical usage. The lack of cellular toxicity in non-neoplastic cells observed in the widespread clinical use of DBS for Parkinson’s disease also gives confidence that adverse effects on the brain will be minimal.

We have demonstrated how both electrical treatments cause significant shifts in cell cycle phase accumulations, with DBS electric fields promoting G0-phase accumulation and TTFields treatments promoting G2/M-phase accumulation in brain tumor cell lines. The accumulation of TTFields treated cells in the G2/M phase has been observed in many different cell lines and tumor types and has been attributed to perturbation of key mitotic proteins, e.g., tubulin and septin proteins [5,18]. A number of mitotic proteins have been investigated; however, many proteins may be influenced by an alternating electric field if there is sufficient current and the protein has a sufficiently high dipole moment. Interestingly, none of the panel of brain tumor cell lines underwent significant levels of apoptosis, which is in line with other reports of TTFields not inducing significant apoptosis in glioma models [27,28]. The mechanisms by which each cell type evades apoptosis is subject to further investigation. The accumulation of DBS electric field-treated cells in the G0-phase is speculated to occur through manipulation of ion flux and its effects on the electrical activity of the cancer cells. Cucullo et al., demonstrated how stimulation of cancer cells with low frequency electric fields may reduce proliferation through an increase in expression of inwards-rectifying potassium channels [29]. An increase in these channels may facilitate greater influx of potassium ions into the cancer cells. It is in part the depletion of intracellular potassium ions which accounts for the depolarized and subsequent proliferative state of the cancer cells [30]. It is also noteworthy that potassium efflux is an essential process to allow progression through the G1/S phase of the cell cycle [30]. Therefore, we propose that the low-frequency DBS electric fields may be potentially influencing ion flux, destabilizing the depolarized membrane potential of the cancer cell and thus promoting G0 phase accumulation of the treated cells.

We have tested a variety of chemotherapies in combination with DBS electric fields, showing that the efficacy of these drugs may be increased with the addition of electric fields treatment. Of particular note, we tested a variety of MGMT-unmethylated and MGMT-methylated GBM cell lines and still observed an increase in drug efficacy in the MGMT-unmethylated cell lines when combining electric fields treatment. This may be of future clinical relevance given how MGMT-unmethylated tumors are typically more chemoresistant to chemotherapies, particularly alkylating agents [31]. TTFields combined with TMZ has shown to increase efficacy, irrespective of MGMT status in vitro [32] and in clinical trials [4]. The combination of electric fields with chemotherapy, especially in certain molecular subgroups, offers a clinical strategy whereby an additional potentially efficacious therapy may be administered without overlapping or significant toxicities [3].

We examined how electric fields affect treated GBM cells at the genetic level to help guide future research and identify novel mechanisms of action. Firstly, we clearly observed striking similarities between the gene expression profiles of DBS electric fields and TTFields treated cells, as with the gene ontologies. A significant number of the affected genes and pathways were associated with mitochondrial and endoplasmic reticulum functioning synonymous with the unfolded protein and the ER stress response. The downregulation of mitochondrial and ER functioning could potentially be a result of the prolonged mitotic arrest from the electrical treatment, with free radical accumulation leading to cellular organelle damage and the corresponding alteration in the gene expression profile. Related observations have already been made in TTFields studies where it has been shown that prolonged TTFields exposure may promote ER stress [8], giving us confidence that the gene expression changes observed are representative of real changes in cellular function. Furthermore, the proposed pathway described ATP (Adenosine Triphosphate) depletion also being a product of TTFields exposure. ATP production through metabolic pathways, such as glycolysis and the citric acid cycle, is reliant upon reduction of NAD^+^ to NADH (Nicotinamide Adenine Dinucleotide) [33]. This redox process of donating and accepting electrons is reliant upon the electron transport chain, and therefore, it may also be potentially a novel mechanism for electric fields treatment in GBM. Such a process of electron movement would be particularly susceptible to changes in the surrounding electrical field such as those generated by TTFields or DBS. However, given the proposed differential mechanisms of action between our electric field treatments, it is possible that DBS electric fields have an underlying function that is yet to be elucidated, which may be addressed with future research. Overall, the gene expression analyses provided novel target genes and pathways which may provide a platform for targeted drug combination studies in the future. Manipulation of the electrical fields surrounding cells gives a credible new approach to oncological therapy that may enhance existing, potentially more toxic therapies with minimal side effects to the normal nervous system.

## 4. Materials and Methods

### 4.1. Cell Culture and Cell Lines

GIN-28, GIN-31 and GCE-77 cells (patient-derived primary low-passage GBM lines) were grown in DMEM (Fisher Scientific, Loughborough, UK) supplemented with 15% FBS. U87-MG (adult GBM) cells were grown in DMEM supplemented with 10% fetal bovine serum (FBS). KNS42 and SF188 (Pediatric GBM) cells were grown in DMEM/F-12 (Fisher Scientific, Loughborough, UK) supplemented with 10% FBS and 1% L-glutamine. DKFZ-EPN1 and BXD-1245-EPN (pediatric ependymoma) cells were grown in DMEM supplemented with 10% FBS. DAOY (SHH medulloblastoma) cells were grown in MEM (MilliporeSigma, St. Louis, MO, USA) supplemented with 10% FBS, 1% L-glutamine, and 1% sodium pyruvate. UW228-3 (SHH medulloblastoma) cells were grown in DMEM/F-12 supplemented with 10% FBS. Neural stem cells and post-mitotic astrocytes were derived from human induced pluripotent stem cells and grown as described by Julia et al., 2017. All cells were maintained at 37 °C in a CO_2_ incubator and have been STR genotyped to ensure correct identification and, in the case of the primary low-passage cell lines, that they maintain the same genetic profile as their parent tumor over multiple passages.

### 4.2. TTFields Application

TTFields were applied using the Inovitro™ system (Novocure, Haifa, Israel). The system consists of two pairs of perpendicular transducer arrays on the outer walls of a ceramic Petri dish. A sinusoidal waveform generator was attached to the transducer arrays to produce alternating electric fields at frequencies between 100–400 kHz and 1.75 V/cm intensity. The electric fields alternated orientation of 90° every second. Temperature was measured to be 37 °C inside the dishes by 2 thermistors attached to the ceramic walls. 

### 4.3. Deep Brain Stimulation Electric Fields Application

DBS electric field treatment was achieved through delivering electric fields from neurostimulators (Medtronic, Dublin, Ireland) via 1 × 8 low impedance compact leads (Medtronic, Dublin, Ireland) and programmed with the N’Vision™ programmer (Medtronic, Dublin, Ireland) with a bipolar configuration with one anode and multiple cathode contacts. The parameters manipulated for the electric field treatment were intensity (V), frequency (Hz) and pulse-width (µs). DBS leads were inserted into T25 flasks containing cells and were programmed to chronically deliver electric fields at the desired parameters.

### 4.4. Intensity and Frequency Titration

Cells were plated and treated for 7 days with DBS electric fields, with cell viability measurements taken at the Day 0, 1, 5 and 7 time-points. A range of intensities were tested: 0 V (sham), 0.5 V, 1.0 V, 2.5 V, 5.0 V and 10.0 V, and a range of frequencies were tested: 60 Hz, 90 Hz, 130 Hz, 160 Hz, 190 Hz, 500 Hz and 1000 Hz.

### 4.5. Cell Viability Assay

PrestoBlue™ was added to each sample at a time-point at a 1:10 dilution. Samples were incubated at 37 °C for 30 min. 100 µL (triplicate) of the PrestoBlue solution was transferred to a black-bottomed 96-well plate, and fluorescence was analyzed via UV-Vis Spectroscopy at excitation/emission wavelengths at 560/590 nm with the FLUOstar Omega (BMG Labtech).

### 4.6. Cell Cycle Analysis

Cells were treated with DBS electric fields for 5 days and then fixed in cold methanol and stored at 4 °C. Samples to be analyzed for cell cycle were incubated in a propidium iodide (PI) solution containing 0.1% Triton X-100, 10 µg/mL PI (Molecular Probes, Inc., Eugene, OR, USA) and 100 µg/mL DNase-free RNase A in cold PBS. The samples were incubated at room temperature in the dark for 30 min. Cell cycle analysis was performed via flow cytometry using FC500 Flow Cytometer (Beckman Coulter, Wycombe, UK) and data were processed with Weasel V.3.1. software.

### 4.7. Chemotherapeutic Combinations

Cells were treated with TMZ (MilliporeSigma, St. Louis, MO, USA), paclitaxel (MilliporeSigma, St. Louis, MO, USA) and mebendazole (MilliporeSigma, St. Louis, MO, USA) as a monotherapy or in combination with DBS electric fields. Each of the drugs were administered at their IC50 (high dose) or 10% IC50 (low dose), with the IC50s being determined through drug-response curves. The drugs were administered with and without DBS electric fields for 5 days before cell viability and cell cycle were assessed.

### 4.8. Gene Expression Analysis

Human Clariom™ S Arrays (ThermoFisher, Waltham, MA, USA) were used for an unbiased, whole transcriptome gene expression analysis of TTFields and DBS-treated GBM cell lines. Microarray processing was performed by Dr. Marcos Castellenos and Dr. Iqbal Khan of Nottingham Arabidopsis Stock Centre (NASC), University of Nottingham.

### 4.9. Gene Expression Data Processing

Hierarchical clustering analyses were performed using Partek^®^. The criteria of significant genes were a false discovery rate (Bonferroni Correction) and *p*-value of <0.05, and a fold-change of <−2.0 and >2.0. An unsupervised clustering method was used, and analyses were displayed as heatmaps on a genome-wide scale within each electrical treatment. Gene ontology and KEGG pathway analyses were performed using Enrichr (Chen et al., 2013, Kuleshov et al., 2016). Enrichr was used at http://amp.pharm.mssm.edu/Enrichr/ (accessed on 1 September 2021).

### 4.10. Western Blot

Cell pellets were collected following electric field treatment and then treated with protease cocktail buffer on ice before centrifugation at 13,000× *g* for 30 min. Protein lysates were then retrieved and quantified via Bradford assay. 20 µg of each protein sample was separated on a 10% acrylamide gel and transferred via electrophoresis to nitrocellulose membranes. The membranes were blocked and then incubated at 4 °C overnight with the primary antibodies: MT-ND5 (Abcam, Cambridge, UK), CTSB (Abcam, Cambridge, UK), BTNL9 (Abcam, Cambridge, UK) and GAPDH 1:10,000 (Abcam, Cambridge, UK). Membranes were washed and then incubated at room temperature for 1 h with complimentary secondary antibodies. Membranes were washed and then incubated in enhanced chemiluminescence and Western blotting detection solutions before Western blot images were developed.

## Figures and Tables

**Figure 1 ijms-23-01982-f001:**
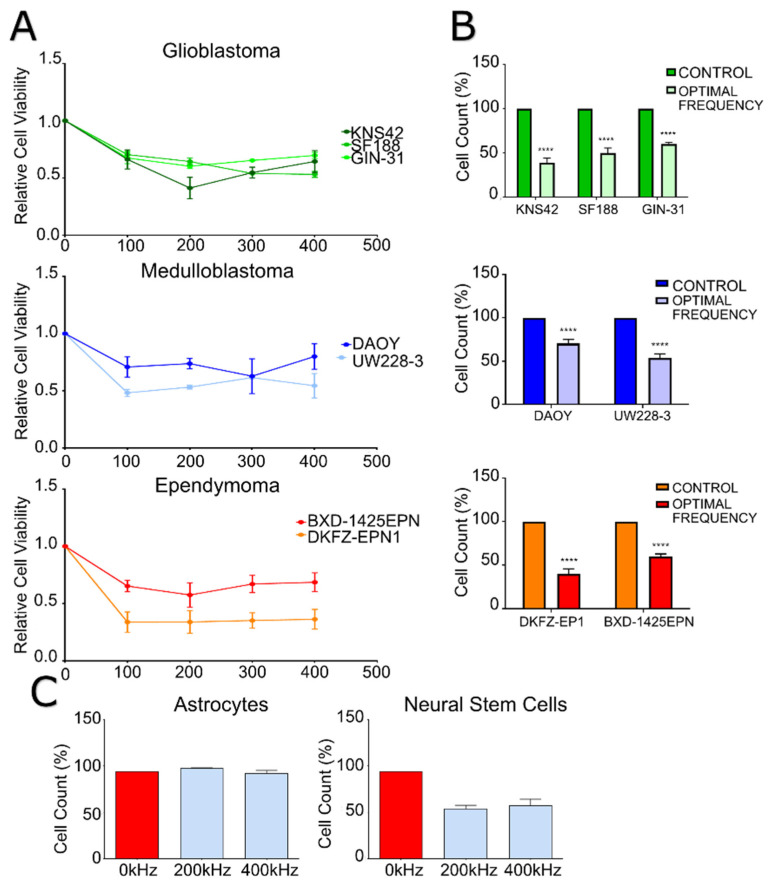
The impact of manipulations of frequency (kHz) on the viability of GBM, medulloblastoma and ependymoma cell lines. (**A**) Cell lines were treated for 72 h over a range of frequencies with the TTFields, with metabolic measurements being taken at the 72 h endpoints. Each cell line demonstrates variable efficacy in TTFields treatment, with no clear pattern of a most optimal frequency for each tumor type tested. (**B**) Cell lines were treated for 72 h at their determined most efficacious frequency (Figure 1A) with the Inovitro, with cell counts being taken at 72 h time-points. Each cell line was significantly affected by TTFields, with variability in efficacy being present between cell lines and tumor types. (**C**) Human neural stem cells and astrocyte cell lines were treated for 72 h at 200 kHz and 400 kHz TTFields with the Inovitro, with cell counts being taken at the 72 h endpoint. The actively dividing neural stem cells were significantly affected by the electric fields, while the non-dividing astrocytes were not affected by either treatment. **** = *p* value < 0.01.

**Figure 2 ijms-23-01982-f002:**
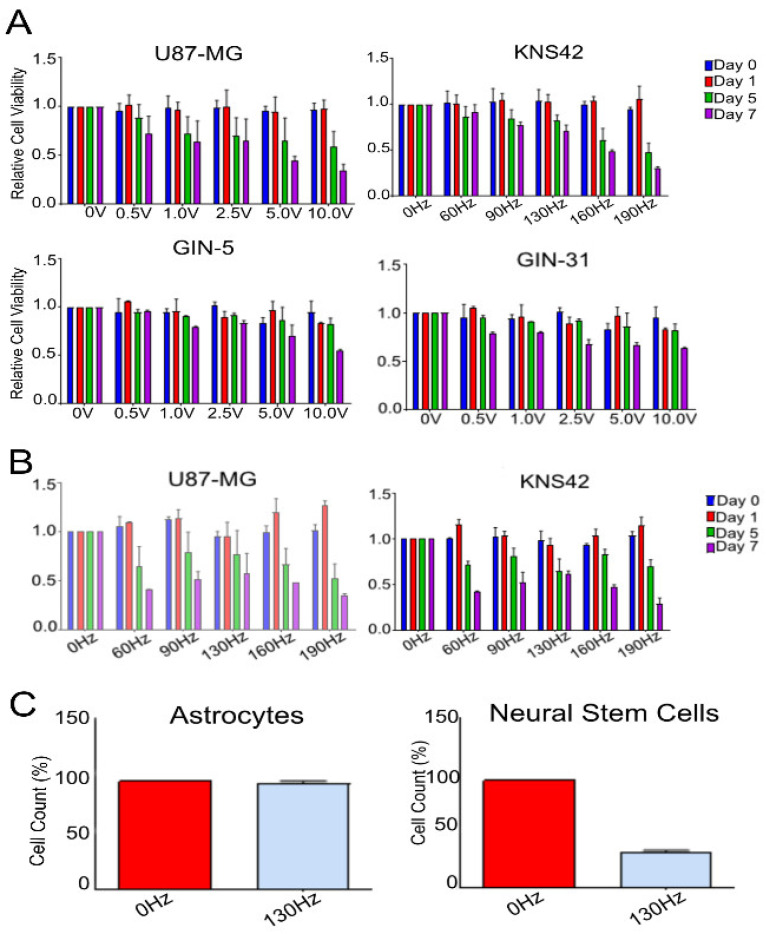
The impact of manipulations of intensity (V) and frequency (Hz) on the viability of commercial and primary GBM cell lines. (**A**) U87-MG, KNS42, GIN-5 and GIN-31 cell lines were treated for 7 days with 130 Hz/450 µs electric fields over a range of intensities, with metabolic measurements being taken at Day 0, 1, 5 and 7 time-points. A clear positive correlation between intensity and duration of treatment is present across all the cell lines tested. (**B**) U87-MG and KNS42 cell lines were treated for 7 days with either 5 V/450 µs electric fields over a range 60–190 Hz or 1 V/450 and 500–1000 Hz frequencies, with metabolic measurements being taken at Day 0, 1, 5 and 7 time-points. Manipulations of frequency between 60–190 Hz demonstrated influence over metabolic activity, while frequencies of 500–1000 Hz had no effect at the tested intensities. (**C**) Human neural stem cells and astrocyte cell lines were treated for 7 days with 10 V/130 Hz/450 µs electric fields, with cell counts being taken at the Day 7 endpoint. The actively dividing neural stem cells were significantly affected by the electric fields, while the non-dividing astrocytes were not affected by the treatment.

**Figure 3 ijms-23-01982-f003:**
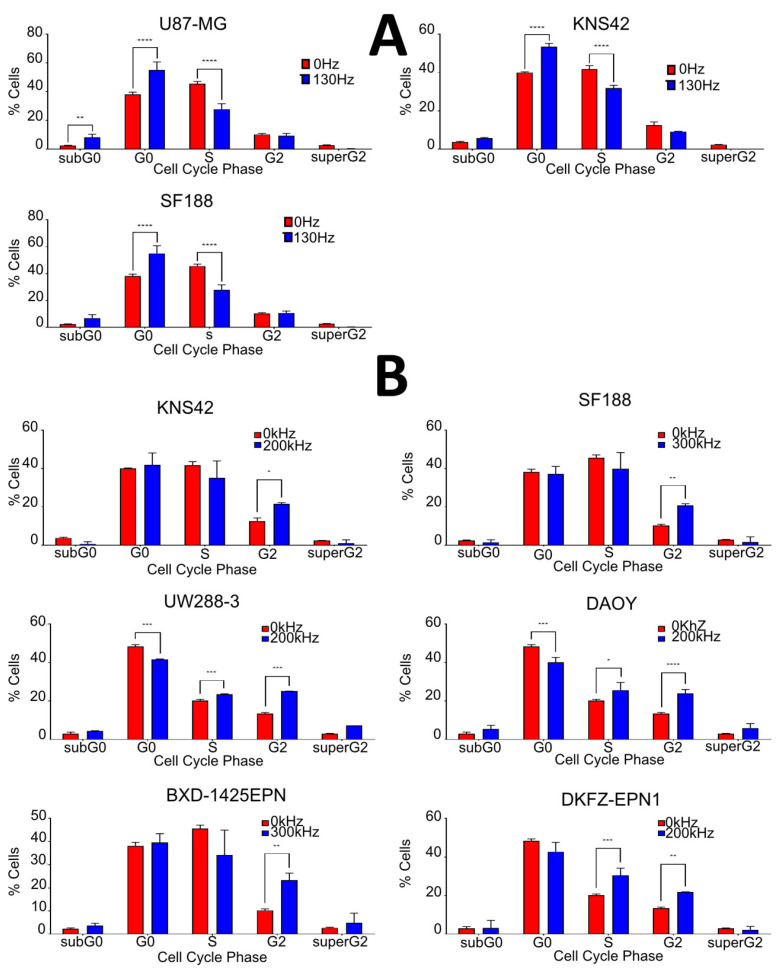
The impact of electric fields on the cell cycle of brain tumor cell lines. (**A**) U87-MG, KNS42 and SF188 cell lines were treated for 5 days with 10 V/130 Hz/450 µs DBS electric fields, with flow cytometry being performed with PI staining to assess cell cycling. The electric field treatments caused significant accumulation of cells in the G0-phase, depletion of cells in the S-phase and minimal accumulation of cells in the subG0-phase. (**B**) Cell lines were treated for 72 h at the determined optimal frequency TTFields (Inovitro), with flow cytometry being performed with PI staining to assess cell cycling. The effects of TTFields treatments were different between cell lines, but the most common response of treatment was significant accumulation of cells in the G2-phase and S-phase and minimal accumulation of cells in the subG0-phase. * = *p* < 0.05 ** = *p* < 0.01 *** = *p* < 0.005 **** = *p* < 0.001.

**Figure 4 ijms-23-01982-f004:**
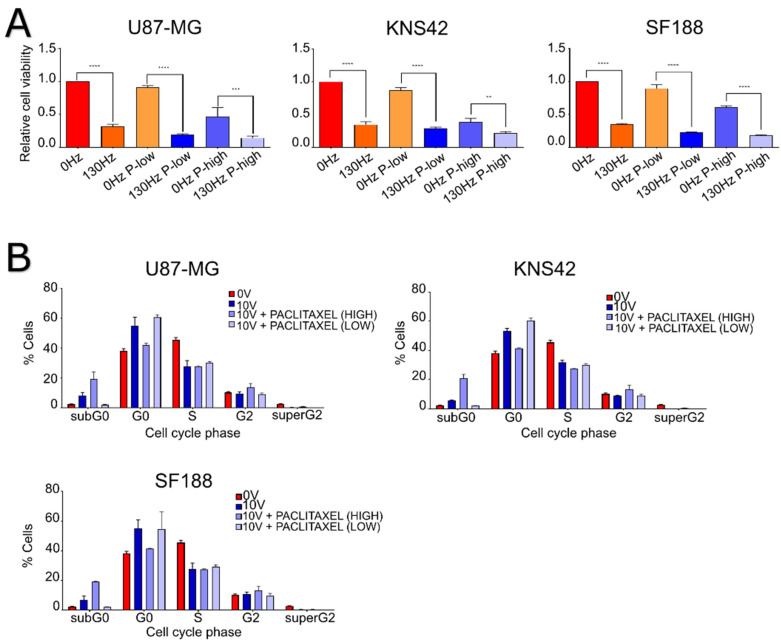
The impact of the combination of DBS electric fields and mitotic inhibitors on the viability and cell cycling of GBM cell lines. (**A**) U87-MG, KNS42 and SF188 cell lines were treated for 5 days with 10 V/130 Hz/450 µs electric fields in combination with a higher and lower dose of paclitaxel, with metabolic measurements being taken at the endpoint. An increase in efficacy of the higher and lower doses of paclitaxel was achieved with the addition of electric fields. (**B**) U87-MG, KNS42 and SF188 cell lines were treated for 5 days with a higher and lower dose paclitaxel, with flow cytometry being performed with PI staining to assess cell cycling. The combination with higher dose paclitaxel caused significant accumulation of cells in the subG0-phase, while the combination with lower dose paclitaxel did not cause differential accumulations of cells relative to electric fields alone. ** = *p* < 0.01 *** = *p* < 0.005 **** = *p* < 0.001.

**Figure 5 ijms-23-01982-f005:**
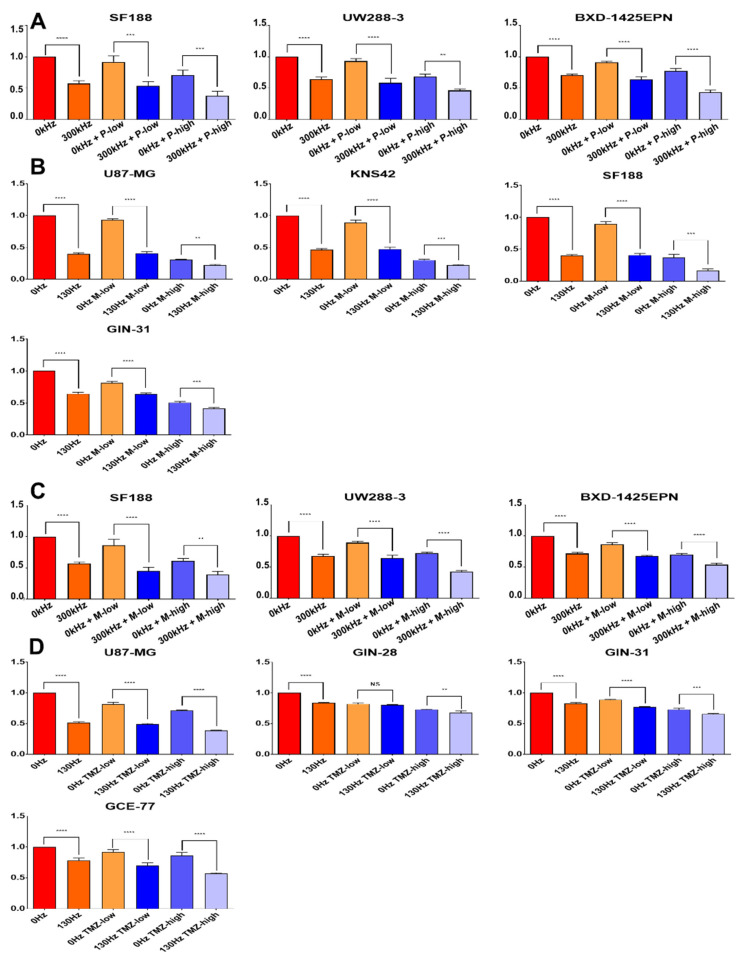
The impact of the combination of DBS electric fields and TTFields with mitotic inhibitors on the viability of brain tumor cell lines. Y axis indicates relative viability compared to untreated cells (0 Hz red column). (**A**) SF188, UW228-3 and BXD-1425EPN cell lines were treated for 3 days with TTFields in combination with a higher (5 nM) and lower (0.5 nM) dose of paclitaxel, with metabolic measurements being taken at the endpoint. (**B**) U87-MG, KNS42, SF188 and GIN-31 cell lines were treated for 5 days with 10 V/130 Hz/450 µs electric fields and a higher (1.25 μM) and lower (0.125 μM) dose mebendazole, with metabolic measurements being taken at the endpoint. (**C**) SF188, UW228-3 and BXD-1425EPN cell lines were treated for 3 days with TTFields in combination with a higher (1.25 μM) and lower (0.25 μM) dose of mebendazole, with metabolic measurements being taken at the endpoint. (**D**) U87-MG, GIN-28, GIN-31 and GCE-77 cell lines were treated for 5 days with 10 V/130 Hz/450 µs electric fields and a higher (5 μM) and lower (0.5 μM) dose TMZ, with metabolic measurements being taken at the endpoint. ** = *p* < 0.01 *** = *p* < 0.005 **** = *p* < 0.001.

**Figure 6 ijms-23-01982-f006:**
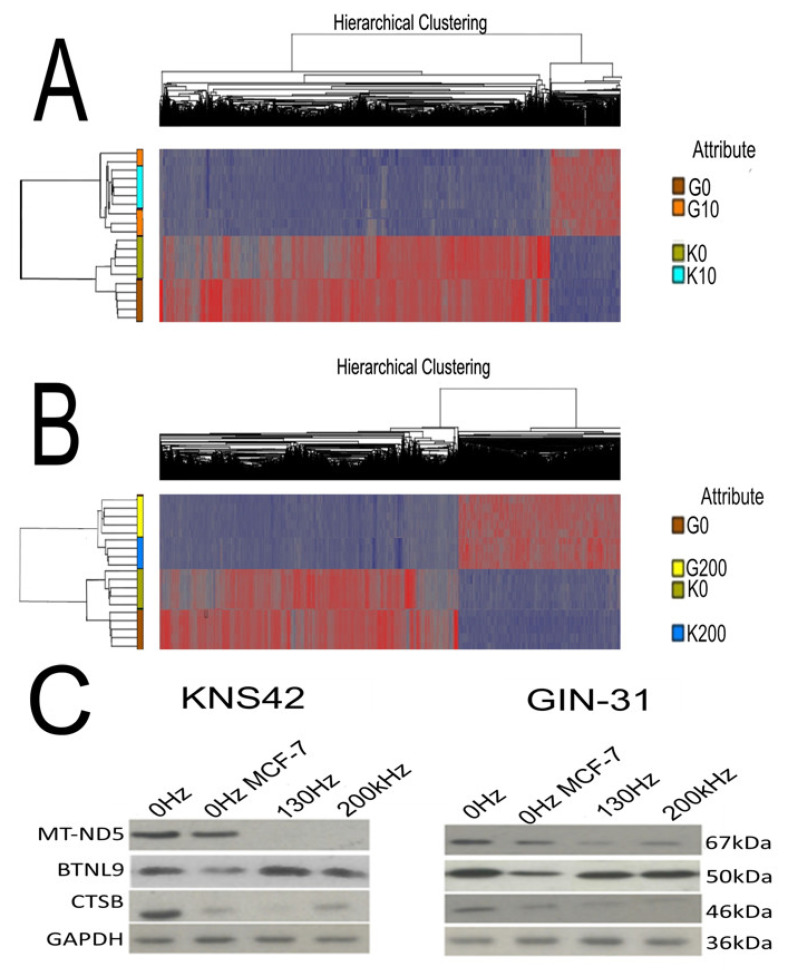
The impact of electric fields on gene expression. (**A**) KNS42 and GIN-31 cell lines were treated for 5 days with 10 V/130 Hz/450 µs electric fields, and gene expression changes were assessed via Clariom™ S Assay arrays (significance criteria: FDR < 0.05 and fold-change <−2 or >2). (**B**) KNS42 and GIN-31 cell lines were treated for 72 h at the determined optimal frequency TTFields (Inovitro), and gene expression changes were assessed via Clariom™ S Assay arrays (significance criteria: FDR < 0.05 and fold-change <−2 or >2). (**C**) Western blot of MT-ND5, BTNL9 and GAPDH as the control. Target genes were among the top 50 most significantly differentially expressed genes for both treatments. The positive controls were breast carcinoma MCF-7 cells and the housekeeping gene chosen was GAPDH. Protein was extracted from samples following 5 days of electrical treatment and 5 days of continuous growth for the untreated samples. Western blot validates downregulation of MT-ND5 and CTSB and the upregulation of BTNL9 following electrical treatment. KEY: G0 = Gin-31 0 Hz sham treated; G10 = GIN-31 130 Hz treated; G200 = GIN-31 200 kHz treated; K0 = KNS42 0 Hz sham treated; K10 = KNS42 130 Hz treated; K200 = KNS42 200 kHz treated.

## Data Availability

Gene expression array data has been deposited at Array Express, accession number E-MTAB-9043.

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
