# Peer review of "Genome-Wide Expression and Anti-Proliferative Effects of Electric Field Therapy on Pediatric and Adult Brain Tumors"

_ijms, 2022, doi:10.3390/ijms23041982_

Round 1

Reviewer 1 Report

The manuscript of original research by Branter et al. titled “Genome wide expression and anti-proliferative effects of electric field therapy on pediatric and adult brain tumors” contains numbers of interesting data related to effectiveness of electric fields therapies (Tumor-treating fields [TTFields] and Deep Brain Stimulation [DBS]) on various brain tumor cell lines. However, the manuscript failed to demonstrate its value because the description is not sophisticated and sometimes inconsistent.

Major comments

  • In “Introduction”, the authors must carefully elaborate both TTF and DBS with similarity and difference between them because the electric fields therapies are not familiar with molecular scientists (main readers of the journal). Moreover, authors have to keep consistency in using special terminology related to the electric field therapies (i.e.; “TTFields therapy”, “Optune”, and “OptuneTTF” for TTFields therapy, “Intratumoral Modulation Therapy (IMT) “, and “Deep Brain Stimulation [DBS]” for DBS, “electrotherapy”, and “electric field therapy” for both) to avoid confusion in understanding by readers. This part contains typos and grammatical mistakes and very uncomfortable to read (see minor comments).
  • In “Results”, authors must simply explain what they did to achieve their research objective with simple description in “Results” because “Materials and Methods” section is located in the last part of the article. Detail description about methods must be located in the “Materials and Methods” section (for example, Presto Blue assay, etc.) but initial appearance of new terminologies should be elaborated in this chapter (for example, Inovitro in the line 106, Medtronic pulse generators in the line 128, PI staining in the line 180, and Clariom S Human Assays in the line 330).
  • The results of this research consist of roughly four parts, (1) antitumor activity of the two different electric fields therapies (sections 2.1 and 2.2), (2) the effect on cell cycle induced either by TTFields therapy or by DBS (I concern that these two are not clearly separated in section 2.3), (3) the effect of electric fields therapies both on antitumor activity (section 2.5) and on cell cycle (section 2.4), and (4) the effect of electric fields therapies on gene expression changes in GBM cell lines (I think this is the highlight of this research and should be explained in more detail). All of them should be systematically constructed (also in “Abstract”, too).
  • In “Discussion”, there are many redundancies which have been already mentioned in “Results” whereas real discussions seem insufficiently described. Because the results of this research consist of four parts as described above, each discussion for each part should be balanced in their volume and depth.

Minor comments

  • In the line 14, an abbreviation for TTFields therapy is described, but the location is not appropriate.
  • Description in the line 15 is not correct because authors themselves explain about it in the “Introduction” and other parts in the manuscript. a short overview of what is known (or not known) will help readers to understand the value of this research.
  • In the line 39, an article is necessary because the subject is single.
  • In the line 50, there is a typo (‘).
  • In the line 55, maybe starting the new paragraph here (from “Other suggested”) instead of starting in the line 57 is better.
  • In the line 56, Intratumoral Modulation Therapy is mentioned, but the similarity or difference with DBS is not clearly described.
  • Between the line 69 and 79, variety of words indicating TTFields therapy are used. Please make a consistency among them.

Author Response

We thank the reviewer for their insightful comments and have made all changes suggested as below:

Point by point responses to reviewer 1 comments: 

Major comments

  • In “Introduction”, the authors must carefully elaborate both TTF and DBS with similarity and difference between them because the electric fields therapies are not familiar with molecular scientists (main readers of the journal). Moreover, authors have to keep consistency in using special terminology related to the electric field therapies (i.e.; “TTFields therapy”, “Optune”, and “OptuneTTF” for TTFields therapy, “Intratumoral Modulation Therapy (IMT) “, and “Deep Brain Stimulation [DBS]” for DBS, “electrotherapy”, and “electric field therapy” for both) to avoid confusion in understanding by readers. This part contains typos and grammatical mistakes and very uncomfortable to read (see minor comments).

- Sentences have been added to the introduction to elaborate the differences between TTFields and DBS for a non-clinical readership which we hope will make this clearer. We have edited the manuscript to try and standardise terminology as much as possible but in this rapidly developing field different terminology is employed by different papers in the literature e.g. IMT and we are forced to use the terminology used by other authors in their studies to some extent. Minor typos and grammar have been corrected.

  • In “Results”, authors must simply explain what they did to achieve their research objective with simple description in “Results” because “Materials and Methods” section is located in the last part of the article. Detail description about methods must be located in the “Materials and Methods” section (for example, Presto Blue assay, etc.) but initial appearance of new terminologies should be elaborated in this chapter (for example, Inovitro in the line 106, Medtronic pulse generators in the line 128, PI staining in the line 180, and Clariom S Human Assays in the line 330).

- We have added a brief explanation/clarification for each technique where it is first mentioned as requested, fuller description remains in the materials and methods section, we are happy to move this into the results section if the editor feels this is required.

  • The results of this research consist of roughly four parts, (1) antitumor activity of the two different electric fields therapies (sections 2.1 and 2.2), (2) the effect on cell cycle induced either by TTFields therapy or by DBS (I concern that these two are not clearly separated in section 2.3), (3) the effect of electric fields therapies both on antitumor activity (section 2.5) and on cell cycle (section 2.4), and (4) the effect of electric fields therapies on gene expression changes in GBM cell lines (I think this is the highlight of this research and should be explained in more detail). All of them should be systematically constructed (also in “Abstract”, too).

- We thank the reviewer for their comment and agree the gene expression data is very interesting, we fully intend to explore that further in future projects. We have clarified section 2.3, hopefully making it clearer that the first paragraph refers to DBS and the second paragraph to TTFields. The results are presented systematically in sections as described and summarised in the abstract which we have amended.

  • In “Discussion”, there are many redundancies which have been already mentioned in “Results” whereas real discussions seem insufficiently described. Because the results of this research consist of four parts as described above, each discussion for each part should be balanced in their volume and depth.

 - The discussion has been extensively revised to remove repetition of results sections and further sentences added to deepen discussion of key elements as requested.

Minor comments

  • In the line 14, an abbreviation for TTFields therapy is described, but the location is not appropriate.

- abbreviation moved to more appropriate location in sentence

  • Description in the line 15 is not correct because authors themselves explain about it in the “Introduction” and other parts in the manuscript. a short overview of what is known (or not known) will help readers to understand the value of this research.

- We have amended the sentence to reflect the reviewer’s concern. There is not space within the word limit of the abstract to describe the various theories regarding the mechanism of TTFields (as described in the introduction).

  • In the line 39, an article is necessary because the subject is single.

-Corrected

  • In the line 50, there is a typo (‘).

-Corrected

  • In the line 55, maybe starting the new paragraph here (from “Other suggested”) instead of starting in the line 57 is better.

-Modified and sentence edited

  • In the line 56, Intratumoral Modulation Therapy is mentioned, but the similarity or difference with DBS is not clearly described.

-Edited and differences described in more detail

  • Between the line 69 and 79, variety of words indicating TTFields therapy are used. Please make a consistency among them

-Terminology has been made more consistent as suggested

Reviewer 2 Report

Glioblastoma multiforme is the most common and lethal brain tumor with a poor median survival of approximately 14 months. The current standard of care, surgical resection followed by radiotherapy plus chemotherapy only provides the patients limited benefits. Recently it is reported an addition of tumor-treating fields to temozolomide chemotherapy could improve progression-free survival of GBM patients. In this study, Branter et al revealed the combination of DBS/TIFE with mitotic inhibitor will increase the efficacy, especially after optimization of frequency or intensity. The authors also monitored the gene expression after treatment of electrotherapy. Generally, the study is well designed, and the manuscript is well-written. There are several concerns listed as follows to be addressed.

  1. The authors claimed that there are interactions between electrical treatments with mitochondrial functions, promoting endoplasmic reticulum stress. The only supporting data is the altered RNA level and protein expressions. It would be much helpful if the authors could conduct some assays to monitor the mitochondrial functions.

  1. Figure 3A and 3B label is missing.

  1. Figure 5, Please indicate the exact concentration of the high dose and low dose of TMZ, paclitaxel and mebendazole. The title of the y-axis is missing.

  1. Figure 6. What is the point to use MCF7 without electric fields treatment as a control?

The expression of MT-NDS and BTNL9 in MCF7 cell lines is not very consistent in the 2 panels. Could the authors clarify it?

  1. CDK1 and CDC25 would be phosphorylated and cyclin B expression is accumulated during G2/M-phase. Could the authors monitor these events after TTFields treatments?

Author Response

Many thanks to the reviewer for their helpful comments which we address as follows:

  1. The authors claimed that there are interactions between electrical treatments with mitochondrial functions, promoting endoplasmic reticulum stress. The only supporting data is the altered RNA level and protein expressions. It would be much helpful if the authors could conduct some assays to monitor the mitochondrial functions.

-We thank the reviewer for their insight and suggestion. The Presto Blue assay of cell viability is a measure of the reductive capacity of a cell which is largely generated by the viability of mitochondria within the cell, we will make the text clearer to emphasise this. We agree further in depth study of mitochondrial function would be very interesting and we will pursue this in future studies. The pattern of gene expression alteration seen after electrical field treatment is characteristic of ER and mitochondrial stress, which is what the comments in the text refer to. 

  1. Figure 3A and 3B label is missing.

 -This has been corrected

  1. Figure 5, Please indicate the exact concentration of the high dose and low dose of TMZ, paclitaxel and mebendazole. The title of the y-axis is missing.

 -Corrected and edited in figure legend

  1. Figure 6. What is the point to use MCF7 without electric fields treatment as a control?

-MCF7 was used as a cell line that is reported already as expressing MT-ND5 and BTNL9 at reasonable levels, whereas this data was not available for KNS42 or GIN31. We sought to show that the Western blot was replicating known data for a previously studied cell line.

The expression of MT-NDS and BTNL9 in MCF7 cell lines is not very consistent in the 2 panels. Could the authors clarify it?

-These are different protein extracts from different biological replicates of MCF7 and we apologise for any inconsistency, though the aim is simply to demonstrate that the blot shows protein expression of these at a recordable level as expected and is not used to draw any further conlcusions.

  1. CDK1 and CDC25 would be phosphorylated and cyclin B expression is accumulated during G2/M-phase. Could the authors monitor these events after TTFields treatments?

-We agree that these would be alternative ways of studying the effects of TTFields on cell cycling and we are grateful to the reviewer for this thoughtful suggestion which we will consider for future studies.

Round 2

Reviewer 2 Report

The authors addressed my major concerns by adding discussions and detailed explanations. The revised version of the manuscript appears to be good. It looks ready for publication as far as I can tell.